# Liver-Based Probabilistic Risk Assessment of Exposure to Organophosphate Esters via Dust Ingestion Using a Physiologically Based Toxicokinetic (PBTK) Model

**DOI:** 10.3390/ijerph182312469

**Published:** 2021-11-26

**Authors:** Jiaqi Ding, Wenxin Liu, Hong Zhang, Lingyan Zhu, Lin Zhu, Jianfeng Feng

**Affiliations:** College of Environmental Science and Engineering, Nankai University, Tianjin 300071, China; dingjiaqi9823@163.com (J.D.); wenxinliu1621@163.com (W.L.); lookatzhanghong@163.com (H.Z.); zhuly@nankai.edu.cn (L.Z.)

**Keywords:** organophosphate esters (OPES), in vitro to in vivo extrapolation (IVIVE), physiologically based Toxicokinetic (PBTK) modeling, health risk assessment

## Abstract

Organophosphate esters (OPEs) are widely used and harmful to organisms and human health. Dust ingestion is an important exposure route for OPEs to humans. In this study, by integrating ToxCast high-throughput in vitro assays with in vitro to in vivo extrapolation (IVIVE) via physiologically based Toxicokinetic (PBTK) modeling, we assessed the hepatocyte-based health risk for humans around the world due to exposure to two typical OPEs (TPHP and TDCPP) through the dust ingestion exposure route. Results showed that the health guidance value of TPHP and TCDPP obtained in this study was lower than the value obtained through animal experiments. In addition, probabilistic risk assessment results indicate that populations worldwide are at low risk of exposure to TPHP and TDCPP through dust ingestion due to low estimated daily intakes (EDIs) which are much lower than the reference dose (RfDs) published by the US EPA, except in some regional cases. Most margin of exposure (MOE) ranges of TDCPP for children are less than 100, which indicates a moderately high risk. Researchers should be concerned about exposure to TDCPP in this area. The method proposed in this study is expected to be applied to the health risk assessment of other chemicals.

## 1. Introduction

Due to the acceleration of industrialization and the widespread use of polymer materials, fires occur frequently all over the world, seriously endangering the safety of people’s lives and property. Therefore, people’s awareness of fire safety is increasing, and all kinds of flame retardants (FRs) are produced and applied [1,2]. Organophosphate flame retardants (OPFRs) have been widely used all over the world because of their low price, low toxicity and low smoke [3,4]. As the yield of OPFRs continues to grow rapidly, its application field is also expanding [5]. In the process of production and use, OPFRs are mainly added into the material by doping and mixing rather than chemical bonding, which makes it easier to enter the environment [6]. In addition, because most OPFRs are semi-volatile and can enter the atmosphere through volatilization, OPFRs are widely distributed in the air and dust, and the residual amount is increasing day by day [7,8]. The molecular structure of organophosphate esters (OPEs) consists of a phosphate skeleton and three substituent groups. According to different substituents, the more common OPEs can be divided into three categories: halogenated alkyl phosphate ester, alkyl phosphate ester and aryl phosphate ester. Different substitutive groups cause great differences in physical and chemical properties of OPEs. For example, with the increase of molecular weight, lg*K*_ow_ of OPEs also tends to increase, but water solubility and vapor pressure decrease accordingly. The larger the molecular weight of OPEs, the weaker the polarity, the less soluble in water, and the less volatile. [9] The main widely used OPEs include triphenyl phosphate (TPHP) and tris (1-dichloro-2-propyl) phosphate (TDCPP) [3]. The wide application of OPEs has gradually brought new risks [6,10]. Studies in many countries have found that OPEs can be detected in almost all indoor dust, with very high concentrations of TDCPP and TPHP [11]. For example, the concentration of TPHP in dust samples from Boston was 7360 ng/g [12].

In recent years, the toxic effect and mechanisms of OPEs have been reported. One study showed that TDCPP is the most harmful OPE to zebrafish [13]. TDCPP significantly upregulated the expression of several biomarker genes (GCK, GSR and NQO1) of liver toxicity in zebrafish [14]. It can cause liver damage [15,16]. OPEs can also affect the life activities of mammals. Studies have shown that high doses of TDCPP can promote an increase of liver weight in rats [17]. TPHP significantly increased the expression of p53 gene in human embryonic liver L02 cells, thereby affecting cell apoptosis [18]. Reported health effects in humans include sick building syndrome from Tri-n-butyl phosphate (TnBP) [19], contact dermatitis, reduction of sperm counts and inhibition of androgen receptors from TPHP, and reduced thyroid hormone levels from TDCPP [20,21].

In recent years, OPEs have been detected in various environmental media such as drinking water, food, indoor air and dust, which means that humans may be exposed to OPEs through ingestion, inhalation and derma absorption [22,23]. Many studies have reported the occurrence of OPEs in indoor dust, which is a crucial daily exposure source of OPEs for humans because OPEs in dust can enter the body via inhalation of dust particles, inadvertent ingestion after hand-to-mouth contact, or direct absorption through the skin [24,25,26,27,28]. Previous studies have reported on the health risk of human exposure to OPEs in the dust [27,29,30,31,32,33]. However, almost all the studies applied the traditional health risk approach published by the EPA based on external exposure dose and reference dose (RfD) [34,35]. First, they measured the concentrations of OPEs in indoor dust in different areas. The external exposure dose of humans was then calculated according to measured concentrations and EPA’s dust exposure formula. Finally, health risk was quantified using the calculated external exposure dose and RfD published by the EPA [34,36,37]. However, there are uncertainties with the existing risk assessment methods. Due to the significant differences between individuals, the results are highly uncertain. In addition, most of the RfDs are calculated from the results of animal experiments, and their accuracy is not high. At present, the traditional risk assessment of OPEs is still based on a small spatial scale or a specific population, which is not extensive. Risk assessment of different people worldwide can help the use and regulation of organophosphate esters. Given the above problems, combining in vitro toxicity data, in vitro to in vivo extrapolation (IVIVE), and margin of exposure (MOE) methods to characterize the safety of chemicals has been proposed as a new approach to risk assessment for world populations [38,39,40]. IVIVE refers to a method of relating in vitro effects to in vivo responses [39,41,42]. It can be broadly defined as an approach utilizing in vitro experimental data to predict in vivo phenomena, including concentrations (e.g., toxicokinetics [TK]) and effects (e.g., toxicodynamics) [38]. Several studies have used the IVIVE method. For example, a study in Taiwan calculated human equivalent doses (HEDs) of bisphenol A and its analogues using the IVIVE process based on the PBTK model and performed risk assessments for different populations [39]. In vitro toxicity data improve the relevance of testing for predicting human responses and provide information for evaluating toxicity pathway perturbations [39]. The physiologically based toxicokinetic (PBTK) model is an essential tool for the IVIVE process, describing the human body as a group of connected compartments [40,43,44]. It can combine an in vitro dose with an in vivo dose. The IVIVE method has been applied to convert in vitro bioactivity concentration from the ToxCast assays to HED [41,42,45]. The MOE has been used by risk assessors as a tool to evaluate safety concerns arising from genotoxic and carcinogenic substances in food and feed, according to the European Food Safety Authority (EFSA) [46].

In this study, we used the EPA in vitro cell-based liver assays data (AC_50_) to evaluate the health risk of OPEs. We applied the PBTK model in combination with a Monte Carlo virtual population to obtain HEDs of different populations (children and adults). Finally, we performed exposure analysis of OPEs through the dust exposure route and risk assessment for different populations in the world.

## 2. Materials and Methods

The workflow of this study is shown in Figure 1. The details of each step are described below. The highlight of this study was the utilization of in vitro data on human hepatocyte assays. We used the activity concentration causing 50% maximum hepatocyte activity (AC_50_) as a point of departure (POD).

### 2.1. Data Collection

#### 2.1.1. Toxicity Endpoint

The liver is an important metabolic organ of the organism that not only regulates the metabolism of substances, but also has important functions of protein synthesis, lipid storage and bile secretion. In addition, as described in Section 1, two OPEs concerned in this study can cause damage to the liver of organisms. Therefore, it is important to evaluate the risk of OPEs to the human liver using hepatocellular toxicity data [18]. In vitro data on human hepatocellular toxicity from the EPA used in our study makes our health risk assessment more robust than previous risk assessments based on animal data.

The ToxCast/Tox21 program is a high-throughput screening program conducted by the U.S. Environmental Protection Agency (EPA), which identified endpoint activity data for more than 1000 in vitro analyses of more than 4000 compounds [39,47,48]. The program investigated different human tissues and organs to provide reliable information for assessing the health risks of chemicals. Two OPEs, TDCPP and TPHP, representative compounds of halogenated alkyl phosphate and aryl phosphate, respectively, are found in high concentrations in dust and have been shown to cause liver damage in living organisms. Therefore, we chose TDCPP and TPHP for health risk assessment. For the derivation of HEDs, the in vitro AC_50_ values of TPHP and TDCPP for each assay were extracted from concentration-response curves from the U.S. EPA’s Chemistry Dashboard (https://comptox.epa.gov/dashboard, accessed on 25 August 2021), which contain the cytotoxicity limit. Due to the potential confounding effect of cytotoxicity [39], not all hepatocyte assays are useful. In this study, we focus on hepatocyte assays having an AC_50_ value below the cytotoxicity limit and four human cell-based liver assays for TPHP and TDCPP were selected. All information about AC_50_ can be found in the U.S. EPA’s Chemistry Dashboard.

#### 2.1.2. Population-Specific Estimated Daily Intakes (EDI) of TPHP and TDCPP through Dust Ingestion

A worldwide literature search was conducted on population exposure to TPHP and TDCPP. A search with “dust” and “organophosphate esters” resulted in about 200 publications in the National Center for Biotechnology Information database PubMed (https://www.ncbi.nlm.nih.gov/pubmed/, accessed on 24 July 2021) and Web of Science (www.webofknowledge.com/, accessed on 15 June 2021). The publication dates were further filtered by the year 2000 to 2021. Studies that include less than five dust samples were excluded from further analysis. It was found that the most important exposure route of organophosphate was dust ingestion. For each literature retrieved, we used EDI medians with high dust ingestion rate (if available) to prevent us from underestimating the risk. The study population was divided into adults (18–79) and children (0–18) so that we could evaluate the risk to different age groups and analyze the differences. The population came from all over the world, so we could analyze the differences in human exposure to TPHP and TDCPP in different regions.

### 2.2. PBTK Model

In this study, the human PBTK model in the high-throughput toxicokinetics (*Httk*) package of R software was applied [49]. The package is available from the Comprehensive R Archive Network (CRAN) r project (https://cran.r-project.org/web/packages/httk, accessed on 5 April 2021). The model was parameterized using high-throughput in vitro data (the intrinsic hepatic clearance, Clint and the plasma protein binding, fub.), as well as human physiological data. *Httk* contains physical and chemical information about hundreds of chemicals from the ToxCast project. Through parameter setting, we can acquire a specific human PBTK model of OPE for the IVIVE process. The model consists of seven compartments: gut, liver, lungs, arteries, veins, kidneys and a single compartment termed “rest”, which was consolidated with unused tissues. *Httk* provides tools for Monte Carlo sampling and reverse dosimetry along with functions that solve concentration vs. time curves, steady state concentrations, the number of days to steady state, and other toxicokinetic summary statistics for chemicals. The parameters of the model included in vitro chemical data, physicochemical properties data of chemicals, physiological and tissue data, and demographic survey data. All data used to parametrize the *Httk* model (physiological, tissue, as well as physicochemical data sources), model structure and model equations are described in previous study [49].

### 2.3. Estimation the Steady-State Concentration with PBTK Model

Population-based liver steady-state concentrations (C_ss_) are predicted using the PBTK model. In this process, we applied the function of Monte Carlo virtual population in *Httk* to consider the individual variability between different ages. Using this function, a virtual population with physiological data taken from the National Health and Nutrition Examination Survey (NHANES) can be generated. The sex, age range, weight category, kidney function category and ethnicity can be defined and together they address the inter-individual variability through the Monte Carlo simulated population. The characteristics of a created population may then be used to generate population-specific parameters to run the PBTK model. Both child and adult populations were used in this study, ultimately to obtain a children-based and adults-based HED for risk assessment. The PBPK model was applied to estimate the population-level C_ss_ of TPHP and TDCPP in the liver of children and adults exposed to a daily dose of 1 mg/kg/day.

### 2.4. Estimation of Human Equivalent Doses Using IVIVE

In vitro hepatocyte activity assays data obtained from ToxCast/Tox21, AC_50_, were converted to the associated external exposure levels of TPHP and TDCPP, HED. The HEDs were the human exposure doses that would be expected to produce C_ss_ of TPHP and TDCPP in the liver of children and adults equivalent to in vitro AC_50_. In this process, the reverse dosimetry was applied to the PBTK model, linking liver concentration with external exposure dose to eliminate the uncertainty generated by animal experiments and highlights the toxicity of the chemicals to the liver. For each OPE, we used AC_50_ values of four in vitro assays of hepatocytes and the population-level C_ss_ in the livers of children and adults to calculate HEDs. The HEDs for the overall hepatocyte effect among all assays combined were obtained by merging HEDs of each assay. The calculation formula is as follows [39,42]:(1)$$\mathrm{HED}\left(\mathrm{mg}/\mathrm{kg}/\mathrm{day}\right)={\mathrm{AC}}_{50}\left(\mu M\right)\times \frac{1\mathrm{mg}/\mathrm{kg}/\mathrm{day}}{\mathrm{Css}\left(\mu M\right)}$$

Different percentiles of HED were calculated. The derived HEDs and EDIs were applied in the following health risk assessment. In addition, the U.S. EPA’s dosimetric adjustment factor (DAF) was used to convert the animal POD (i.e., NOAEL or LOAEL) values to the HEDs using Equation (2) to compare the results of our study [39,42].
(2)$$\mathrm{HED}\left(\mathrm{mg}/\mathrm{kg}/\mathrm{day}\right)=\mathrm{Animal}\;\mathrm{POD}\left(\mathrm{mg}/\mathrm{kg}/\mathrm{day}\right)\times \mathrm{DAF}$$

### 2.5. Risk Assessment

In this study, HEDs for TDCPP and TPHP was applied to evaluate the human health risk for different populations in different countries worldwide through the MOE approach [50,51]. The population-specific estimated daily intakes (EDI) of TPHP and TDCPP available from the world were compared with the HED to calculate the MOE according to the following formula [39,52].
(3)$$\mathrm{MOE}=\frac{\mathrm{HED}}{\mathrm{EDI}}$$

EDI medians with a high dust ingestion rate of each population and different percentiles of HED were used to perform the probabilistic risk assessment. The MOE evaluates the risk into three levels of concern: high concern (MOE ≤ 1), moderate concern (1 < MOE < 100) and low concern (MOE ≥ 100) [53]. In the following section, we analyzed the risk characterization based on MOE of TPHP and TDCPP for children and adults in different regions.

## 3. Results

### 3.1. Hepatocyte-Based Css in Liver, AC50 and HEDs for TPHP and TDCPP

The concentrations of TPHP and TDCPP in the liver for children and adults following daily oral exposures is shown in Figure 2. The C_ss_ values of different percentiles are given (P2.5, P25, P50, P75 and P97.5). The predicted C_ss_ levels of TPHP and TDCPP for children were 8.17 (1.31–139.30) and 25.34 (5.72–241.45) μM, and for adults were 24.1 (3.86–251.97) and 58.66 (14.6–481.66) μM, respectively.

For the two studied OPEs, AC_50_ values for the 4 ToxCast HTS assays related to the liver and calculated HEDs are presented in Table 1 after cytotoxicity filtering. The biological process targets of four assays are all regulation of transcription factor activity. According to Formula (1), in vitro AC_50_ values and the population-level C_ss_ in the liver predicted by the PBTK model were used to calculate the HEDs. This process is termed as the application of a reverse dosimetry approach of the PBTK model. The HEDs in our study are liver-based guidance for assessing the risks to human health. Table 1 also presents the HED estimates associated with the liver derived from each of the in vitro AC_50_ values and the overall HED estimates for assays for TPHP and TDCPP. The HEDs of TPHP associated with overall hepatocyte assays were 0.266 (P2.5–P97.5: 0.01–2.913) and 0.096 (0.005–1.015) mg/kg/day, for children and adults, respectively. HEDs of TDCPP for children and adults were 0.042 (0.003–0.679) and 0.018 (0.001–0.267) mg/kg/day.

HEDs of TPHP and TDCPP derived from in vitro assays for each biological process target and the overall hepatocyte assay combined with box-whisker plots are shown in Figure 3 and Figure 4. The result is presented as different percentiles. Median HEDs of TPHP and TDCPP shown in Figure 4 range from 0.01 to 1 mg/kg/day for both children and adults. For each OPEs, HED levels of children are higher than that of adults in most percentiles. Moreover, a method for calculating HED from animal experimental data has been reported. We selected the smallest available POD value for the extrapolation of HED. Based on a male rat study with a NOAEL of 105 mg/kg/day and a DAF of 0.24, the HED estimate for TPHP is 25.2 mg/kg/day for system effects following 90-day exposure. For TDCPP, the HEDs estimate based on a POD (LOAEL) of 5 mg/kg/day and a DAF of 0.26 from a female rat study was 1.3 mg/kg/day for the renal system effect. The details of animal-based POD of TPHP and TDCPP can be found from the ECHA database (https://echa.europa.eu/, accessed on 25 November 2021).

### 3.2. Population-Specific EDIs of TPHP and TDCPP via Dust Ingestion

In this study, we conducted an extensive literature search to evaluate human exposure to OPEs via dust around the world. The total estimated daily intake of TPHP and TDCPP via dust ingestion is shown in Figure 5. The regions covered include Asia, Africa, Europe, Oceania, North America and South America. The population included children and adults. From each study we extracted the population EDI medians calculated using high dust ingestion parameters for exposure analysis (Table 2).

### 3.3. Risk Assessment of TPHP and TDCPP

This study used AC_50_ values based on hepatocyte assays and EDIs through dust ingestion, so we finally assessed the liver risk of exposure to TPHP and TDCPP in different populations. The distributions of the MOE estimates of TPHP and TDCPP (P2.5–P97.5, P1–P99 and P0.1–P99.9) for each population are shown in Figure 6. The P2.5–P97.5 intervals of MOE estimates of TPHP for all populations fell within the low-risk area, except for the Latvian children with a high EDI. Moreover, a highly conservative assessment using the interval of P0.1–P99.9 showed a low concern for the risk of TPHP for adults surveyed in this study, except for the Latvian adults. For adults in most areas, therefore, TPHP uptake via ingestion of dust posed a very low risk to the liver. Although most of the MOE ranges of TPHP for children and adults fall in low-risk areas, the MOE values of children are generally lower than those of adults, indicating children have a higher liver risk than adults. For TPHP, children in part of the country have MOE at the very lower-end estimate (P0.01), which fell within the yellow zone (moderate concern).

Most of the interval of TDCPP-based MOE for children in Latvia and Nanjing, China, fell within the moderate concern zone. Furthermore, the levels of P1 and P0.1 of MOE even fell in the high concern zone. This result indicated that the presence of TDCPP in dust should arouse attention for children and adults since the MOE of TDCPP was higher than TPHP. Compared to Latvian children exposed to TDCPP, children in Brazil and Saudi Arabia had slightly low MOE scores, of which about half of the interval is in the medium-high risk area. So, children in both countries remain at higher risk due to exposure to TDCPP compared to those in other countries. As a result of exposure to TDCPP through dust intake, the population of children in most of the areas involved in this study, such as China, Korea, Australia, Egypt, South Africa and some European countries, had P0.1 levels of MOE in moderate risk areas. A highly conservative assessment using the interval of P0.1–P99.9 showed a low concern for the liver risks of TDCPP for children in Japan, Pakistan, Romania and Portugal. Hence, it suggests that there was negligible liver risk to children from TDCPP exposure through dust. The P0.1–P99.9 intervals of MOE estimates for European and Chinese adults fell within the low concern zone, except for Latvian and Nanjing adults. Therefore, the liver risk caused by dust exposure to TDCPP in these people is also very low.

## 4. Discussion

For PBTK model simulation results, C_ss_ levels of children are lower than C_ss_ levels of adults in most percentiles. There are two reasons accounting for this phenomenon: (1) The parameters of the Monte Carlo simulation population showed that the metabolic rate of these two OPEs was significantly higher in children than in adults. So, the amount of OPEs remaining in the liver of children is significantly reduced when a steady state is reached. (2) The input value, 1 mg/kg/day, is related to body weight. An absolute value of actual intake for one person will be small if body weight is light. Compared to adults, children weigh less, so their actual chemical intake is lower than that of adults. Furthermore, C_ss_ of TDCPP in the human liver was higher than that of TPHP in most percentiles. The structural and physicochemical properties of TDCPP and TPHP may lead to this result. The information of TPHP and TDCPP derived from the U.S. EPA’s Chemistry Dashboard indicated that the in vitro intrinsic hepatic clearance value of TPHP is higher than that of TDCPP. Higher clearance values may result in faster excretion of chemicals from the human body. By comparing C_ss_ of TPHP and TDCPP, we concluded that TDCPP may be more harmful to human liver tissue than TPHP. TDCPP belongs to halogenated alkyl phosphate containing a substituent of chlorine. Chlorinated OPEs have been shown to be more water soluble, resulting in the possibility of higher concentrations in the human body. TDCPP is classified under regulation EC 1272/2008 as a category 2 carcinogen with hazard statement H351 “suspected of causing cancer” by the EU [25]. Besides, the ranges of in vitro-based HEDs were lower than the animal-based HEDs, which indicates that HEDs calculated in this study are more conservative. Therefore, overall liver-based HEDs were used to conduct a risk assessment of exposure to TPHP and TDCPP.

For EDIs of different populations, children have significantly higher exposure to TPHP and TDCPP via dust ingestion than adults in each country or region. Children are often more heavily exposed to OPEs in the environment than adults because children ingest more dust than adults [24,68]. Children’s behavior patterns, such as playing close to the ground and more frequent hand-to-mouth activity, increase their exposure to potential OPEs via dust [27,28]. In addition, children have lower body weights and spend more time at home compared to adults resulting in greater dust ingestion [61]. In a Latvia study [11], the median values for EDIs of TPHP and TDCPP found in children were 560 and 1570 (ng/kg/day), respectively. The adults’ median EDIs of TPHP and TDCPP were 24 and 67.4 (ng/kg/day), which were about 1/20 of EDIs for children. Particularly high EDIs of TPHP and TDCPP were also found in populations in Nanjing in China [54], with an estimated median value of 39 and 1369 ng/kg/day for children and 2.3 and 51 ng/kg/day for adults, respectively.

High levels of EDIs for children were found in populations in Saudi Arabia [29], Brazil [1] and South Africa [66] than in other studied children. Especially, the median EDI of TDCPP, which is the most important contributor for total EDIs of OPEs, is significantly higher than that of TPHP for each population in Saudi Arabia and Brazil. Low total EDIs of TPHP and TDCPP for children were found in some Asian countries, such as South Korea [58], Pakistan [27], the Philippines [60] and Japan [59], with a range of 2.15–35.8 ng/kg/day. In the study for the Philippines, they excluded TDCPP from their target OPEs due to background contamination problems in the lab. Only the median EDI for TPHP was recorded, so we were unable to evaluate exposure levels for TDCPP. The lowest levels of total EDIs for children were found in the United States (New York) [8], India [12], Columbia [12], Vietnam [12] and Nepal [37], ranging from 0.02–0.1 ng/kg/day. Two studies have evaluated human exposure via dust ingestion and calculated the median EDI of TPHP and TDCPP among different population groups in South Africa and Egypt [25,66]. The results showed that adults and children in Africa had a higher total intake of TPHP and TDCPP through dust than in China (except Nanjing) and some European countries [30,31,33,61,62,67]. The South African report stated that indoor characteristics, such as the existence of electronic products and foams in homes, are important factors affecting the concentrations of OPEs in dust and human ingestion. Results from Egypt revealed that children’s exposure estimates to ΣOPEs (all detected OPEs) via dust ingestion in Egypt are generally in line with those reported from Pakistan. However, the total EDIs of TPHP and TDCPP for adults and children in Egypt were higher than that of Pakistani adults and children because the main OPEs in indoor dust in Pakistan are not just TPHP and TDCPP.

Several studies have reported levels of EDIs of TPHP and TDCPP in populations from different regions of China including Beijing [55], Nanjing, Guangzhou [57], South China [56] and North China [34]. Except for Nanjing, EDIs of the population in other regions are at a low level. A study showed that people in rural areas consume more of these two kinds of OPEs through dust ingestion than people in urban areas of southern China, and TDCPP is the main contributor [56]. Due to poor sanitation in rural areas compared to urban areas, dust ingestion of people in rural areas is much higher than in urban areas, resulting in higher EDIs in the rural population [56]. Rural and urban populations ingest more TPHP than TDCPP. The high EDIs of TPHP via rural indoor dust may originate from e-waste recycling, with TPHP subsequently transported to the surroundings. The differences in the furniture, interior decoration and building materials between the urban and rural areas are likely the main reason for the different EDIs [56]. In southern China, people’s EDIs of TPHP and TDCPP via indoor dust ingestion in college dormitories are close to adults’ EDIs in urban areas [56]. In Guangzhou [57], China, people intake about five times as much TDCPP as TPHP through dust ingestion. In a Beijing study [55], people’s EDIs of TPHP are close to that of TDCPP. A population exposure analysis of TPHP and TDCPP in northern China involved only adults, with an estimated median value of 0.42 and 0.33 ng/kg/day, respectively. Another study of the population based EDIs of two OPEs in Japan involved only children, for which the median EDIs of TPHP and TDCPP were 1.89 and 0.27 ng/kg/day, respectively.

For Oceania, our study only involved adults and children in Australia [26], who’s median EDIs of TPHP and TDCPP through dust ingestion are low compared to the Chinese population. The study in Australia also analyzed the relationship between concentrations of OPEs and indoor products, and the results showed that significantly higher concentrations of TDCPP (*p* < 0.001, medians of 0.93 and 0.26 mg/g in carpeted and uncarpeted houses, respectively) were found in dust collected from carpeted houses. Although the concentrations of OPEs and EDIs are related to some products in the room, whether the causal relationship exists needs to be further investigated. (1) The existence and interaction of multiple OPEs in indoor environments, (2) differences in behavior (such as degradation and air-dust partition) of individual OPEs in environmental matrices because of their specific physical properties, and (3) the limitations of test schemes and statistical methods will affect the results.

In the European countries surveyed in this study, except for Latvia, the people’s EDIs of TPHP and TDCPP through dust ingestion were very low. In a German study [27], median EDIs of TPHP and TDCPP for adults were 0.21 and 0.27 ng/kg/day, and for children were only 0.44 and 1.5 ng/kg/day, respectively. OPEs were evaluated in household dust from Belgium [61], Italy and Spain [62]. Then, data obtained were used to determine EDIs via house dust ingestion for children and adults. Children’s median EDIs of TPHP and TDCPP were 1.96 and 1.97 ng/kg/day, respectively. The overall EDIs of children in these three countries is at the same level as that of Belgian children. Therefore, we infer that TPHP and TDCPP exposure levels, which were low, may be the same for children in these three European countries. Several other studies also reported the exposure levels of TPHP and TDCPP in some other European countries and calculated the EDIs through dust ingestion. For adults, Latvia has the highest total EDIs level, followed by Norway > Belgium > Portugal > Germany > Romania > New Zealand > UK. Latvian children, like adults, have the highest EDI, and the EDI ranking of children in other countries is as follows: Portugal > UK > Romania > New Zealand > Belgium > Germany. Therefore, we can conclude that for the two OPEs, TPHP and TDCPP, there are great differences of EDI levels via dust ingestion among people in different countries. The discrepancy may reflect country-specific applications of different OPFR substances. Moreover, the EDI levels of adults and children may be different in different countries. For example, the EDI of adults in Germany is larger than that of the UK, but the EDI of children is lower than that of the UK.

For the results of risk assessment, we compared the results of this study with those previously reported. In the Saudi Arabia study, EDI was multifold below RfD values for TDCPP and TPHP for exposure from dust [29]. However, estimated incremental lifetime cancer risk (ILCR) with moderate risk (1.5 × 10^−5^) for Saudi adults and calculated hazardous index (HI) of >1 for Saudi children from bis(2-ethylhexyl) phthalate (DEHP) showed a cause of concern to local public health. Our results suggest that TDCPP may pose a risk to both adults and children in Saudi Arabia, as some of the MOE ranges are in the medium-risk zone. It has been shown that the harm of different OPEs to the human body should be paid attention to in Saudi Arabia. Therefore, extensive scale studies will be needed to report on the health risks associated with more OPEs. The median concentration of TPHP in Latvia [6] was similar to that found in household environments in Brazil. The health risks associated with TPHP were also at a low level for both adults and children, consistent with our findings. However, a previous study has suggested that TDCPP in dust poses no risk to Latvian children, which contradicts our findings. In the Egypt study, maximum exposure estimates for adults and toddlers via indoor dust ingestion were much lower than the reported reference doses. Collectively, the results of this study indicate that current human exposure to TDCPP and TPHO via accidental ingestion of indoor dust does not pose an immediate health risk to the Egyptian population based on the current state of knowledge about the toxicological properties of OPEs [25]. The results of our study are similar to those of the Egypt study, with low levels of risk for both adults and children exposed to TDCPP and TPHP. According to a study in Nanjing, China, EDIs for adults and children to TDCPP and TPHP via dust ingestion were much lower than the RfDs. Therefore, previous researchers concluded that there was no risk of exposure. However, the results of this study indicate that children are at a higher risk from exposure to TDCPP through dust, which differs from previous results.

The health risk assessment of OPEs is challenging due to the uncertainties of traditional risk assessment and the lack of global surveys. In the present study, we addressed this challenge by converting in vitro ToxCast hepatocyte assays data from EPA into HEDs that could be applied in an MOE risk assessment approach using population based IVIVE via PBTK model, and by integrating EDI of OPEs in different populations worldwide. We used HEDs deriving from IVIVE for risk assessment, which gives us a better understanding of the risks posed by TDCPP and TPHP in dust. However, there are some limitations to this study. The EDIs obtained from the literature we surveyed were not conducted at the same time, so our risk assessment may lack timeliness for some countries, but this cannot be avoided. In addition, we applied the PBTK model of the human body in the *H**ttk* package for health risk assessment, but the PBTK model is a high-throughput model that can be adapted to a variety of compounds by changing the parameters. Therefore, its description of ADME of both OPEs may be inadequate. We will develop PBTK models specific to TDCPP and TCEP in future studies.

## 5. Conclusions

In this study, by integrating ToxCast high-throughput in vitro assays with IVIVE via PBTK modeling, we assessed hepatocyte-based health risk for people around the world due to exposure to TPHP and TDCPP through dust ingestion. We have reduced the uncertainty of the risk assessment approach reported from EPA to a certain extent. The HED values for TPHP and TDCPP derived from ToxCast in vitro assays are more conservative than the HED obtained from animal studies. This can lead us to be more cautious about the health risks posed by TPHP and TDCPP. In addition, we applied a Monte Carlo virtual population technique to quantify the population differences and uncertainties in risk assessment in a probabilistic way. For adults and children in most areas, exposure to TPHP and TDCPP through ingestion of dust has not yet reached alarming levels. However, due to the wide application of OPEs in various fields, more data are needed of the exposure through other pathways in future research.

## Figures and Tables

**Figure 1 ijerph-18-12469-f001:**
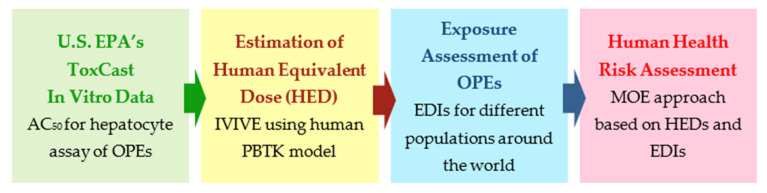
Workflow for risk assessment.

**Figure 2 ijerph-18-12469-f002:**
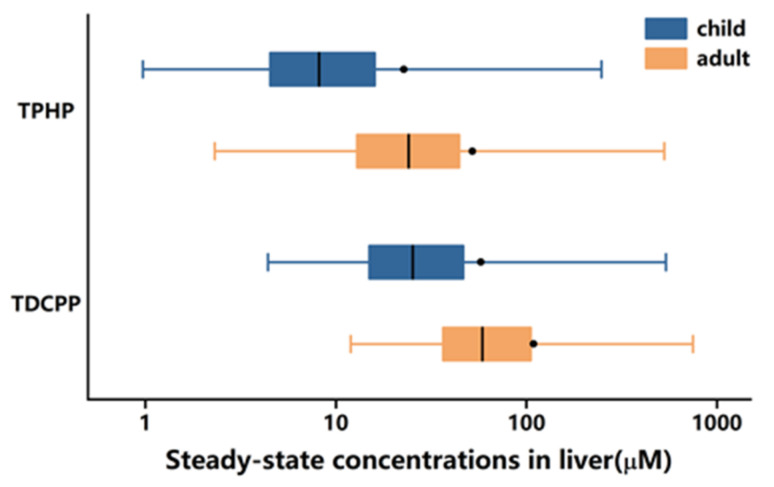
Predicted C_ss_ of TPHP and TDCPP in the liver of children and adults following a daily dose of 1 mg/kg/day. C_ss_ derived from PBTK simulation are presented with box-whisker plots.

**Figure 3 ijerph-18-12469-f003:**
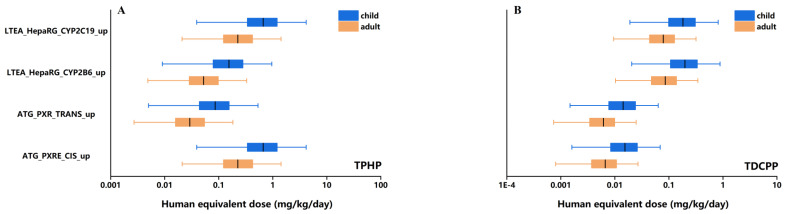
HEDs of TPHP (**A**) and TDCPP (**B**) derived from in vitro hepatocyte assays. Different percentiles of HED for each population were calculated (P2.5, P25, P50, P75 and P97.5).

**Figure 4 ijerph-18-12469-f004:**
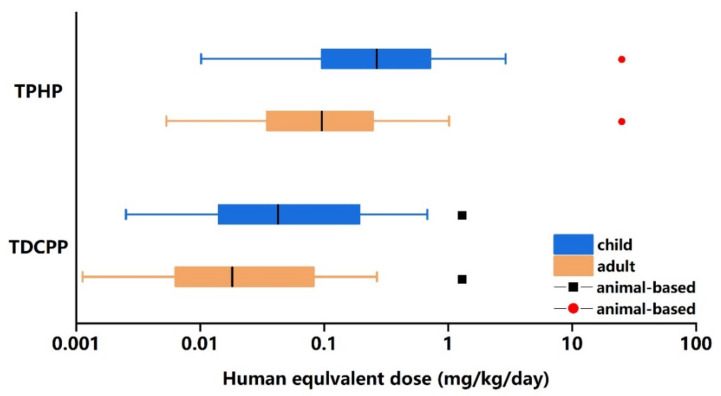
Box and whisker plots represent overall liver based HEDs of TPHP and TDCPP obtained by merging HEDs of each assay. Different percentiles of HED for each population were calculated (P2.5, P25, P50, P75 and P97.5). Black squares and red dots represent HED values derived from experiments on animals.

**Figure 5 ijerph-18-12469-f005:**
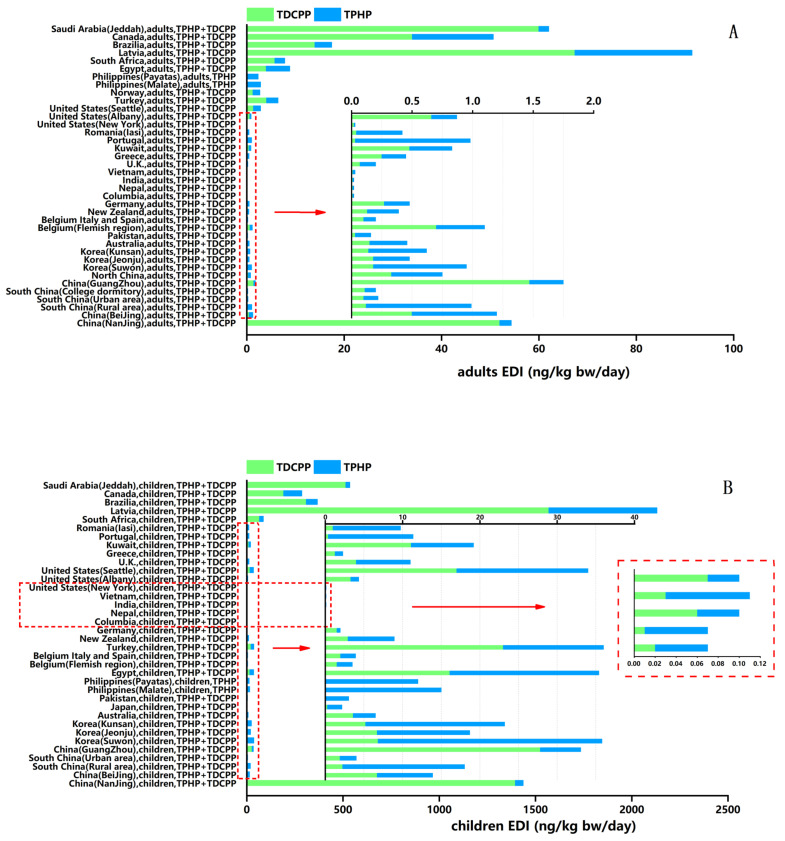
Total estimated daily intake of TPHP and TDCPP for adults (**A**) and children (**B**). The descriptions in the Y-axis show the following information: countries/regions, population characteristics and the types of OPEs. The X-axis of the two graphs have different ranges.

**Figure 6 ijerph-18-12469-f006:**
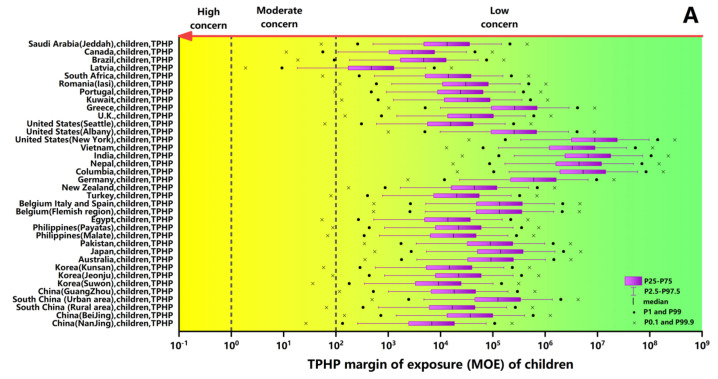
Health risk assessment for children and adults of exposure to TPHP (**A**,**B**) and TDCPP (**C**,**D**) via dust ingestion around the world. The descriptions in the Y-axis show the following information: countries/regions, population characteristics and the types of OPEs. The descriptions in the X-axis show the scope of the MOE.

**Table 1 ijerph-18-12469-t001:** AC_50_ values and the corresponding estimated HED values for TPHP and TDCPP.

Assay Name	TDCPPAC_50_ ^1^	Adult Population HED ^2^	Child Population HED	TPHP AC_50_	Adult Population HED	Child Population HED
LTEA_HepaRG_CYP2B6_up	5.01	0.085 (0.01~0.343)	0.198 (0.02~0.876)	1.26	0.052 (0.005~0.327)	0.154 (0.009~0.961)
LTEA_HepaRG_CYP2C19_up	4.62	0.079 (0.01~0.316)	0.182 (0.019~0.808)	5.43	0.225 (0.022~1.409)	0.665 (0.039~4.140)
ATG_PXRE_CIS_up	0.39	0.007 (0~0.027)	0.015 (0.002~0.068)	5.44	0.0226 (0.022~1.411)	0.667 (0.039~4.148)
ATG_PXR_TRANS_up	0.36	0.006 (0~0.025)	0.014 (0.001~0.063)	0.7	0.029 (0.003~0.182)	0.086 (0.005~0.534)
Overall liver assays	-	0.018 (0.001~0.267)	0.042 (0.003~0.679)	-	0.096 (0.005~1.015)	0.266 (0.01~2.913)

Note: The units of the AC_50_ and in vitro-based HED values are μM and mg/kg/day, respectively. AC_50_, activity concentration causing 50% maximum hepatocyte activity; TPHP, triphenyl phosphate; TDCPP, tris (1-dichloro-2-propyl) phosphate; C_ss_, steady-state concentrations in liver; HED, human equivalent dose. ^1^ AC50 values were obtained from concentration-response curves in the U.S. EPA’s Chemistry Dashboard. ^2^ HED was estimated by AC_50_ × 1 (mg/kg/day)/C_ss_. Values are presented as the 50th percentile (2.5th–97.5th percentiles).

**Table 2 ijerph-18-12469-t002:** EDIs (ng/kg-bw/day) of TPHP and TDCPP via dust ingestion for children and adults from different regions.

Region	TPHP		TDCPP		Reference
	Adults	Children	Adults	Children	
China (Nanjing)	2.3	39	51	1396	[54]
China (Beijing)	0.7	7.2	0.5	6.7	[55]
South China (rural area)	0.87	15.8	0.12	2.23	[56]
South China (urban area)	0.12	2.12	0.1	1.89	[56]
South China (college dormitory)	0.09	-	0.11	-	[56]
China (Guangzhou)	0.28	5.25	1.47	27.81	[57]
North China	0.42	-	0.33	-	[34]
Korea (Suwon)	0.77	29	0.18	6.8	[58]
Korea (Jeonju)	0.3	12	0.18	6.7	[58]
Korea (Kunsan)	0.48	18	0.14	5.2	[58]
Australia	0.37	2.9	0.15	3.6	[26]
Japan	-	1.89	-	0.27	[59]
Pakistan	0.13	2.97	0.03	0.06	[27]
Philippines (Payatas)	2.3	12	-	-	[60]
Philippines (Malate)	2.8	15	-	-	[60]
Egypt	4.8	19.3	4	16.1	[25]
Belgium (Flemish)	0.4	2	0.7	1.5	[61]
Belgium, Italy and Spain	0.1	1.96	0.1	1.97	[62]
Turkey	2.3	13	4.1	23	[63]
New Zealand	0.26	5.99	0.13	2.93	[30]
Germany	0.21	0.44	0.27	1.5	[32]
Columbia	0.01	0.05	0.01	0.02	[12]
Nepal	0.01	0.06	0.01	0.01	[37]
India	0.01	0.04	0.01	0.06	[12]
Vietnam	0.02	0.08	0.01	0.03	[12]
United States (New York)	0.01	0.03	0.02	0.07	[8]
United States (Albany)	0.21	1.04	0.66	3.28	[12]
United States (Seattle)	1.4	17	1.4	17	[64]
UK	0.13	7	0.07	4	[31]
Greece	0.2	1.02	0.25	1.25	[12]
Kuwait	0.35	8.1	0.48	11.09	[27]
Portugal	0.95	11	0.03	0.37	[33]
Romania	0.38	8.75	0.04	0.98	[65]
South Africa	1.97	18.75	5.8	66.85	[66]
Latvia	24	560	67.4	1570	[11]
Brazil	56	3.4	310	14	[1]
Canada	16.6	93	34	192	[63]
Saudi Arabia	2	20	60	515	[29]
Norway	1.36	-	1.3	-	[67]

## Data Availability

Data is contained within the article.

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
