# Peer review of "Liver-Based Probabilistic Risk Assessment of Exposure to Organophosphate Esters via Dust Ingestion Using a Physiologically Based Toxicokinetic (PBTK) Model"

_ijerph, 2021, doi:10.3390/ijerph182312469_

Round 1

Reviewer 1 Report

The purpose of the study is to assess hepatocyte-based health risk for different populations in different countries due to exposure to two typical organophosphate-eating OPEs (TPHP and TDCPP) via the dust ingestion exposure route. Specifically, the authors used data from EPA in vitro liver assays to study the potential health effects of OPEs, applying the physiology-based toxicokinetic model in combination with a virtual Monte Carlo population to obtain the equivalent human doses of adult and child subjects (age less than 18 years)

The study design can be considered of considerable originality, and using innovative techniques for risk assessment studies. Furthermore, the topic is of great relevance to the scientific community because of ubiquitous population exposure to these compounds. However, there are some critical issues in the manuscript, and more clarity in the exposition of the concepts could be helpful to improve the quality of the article.

Suggestions for authors.

Abstract, lines 15-16: The authors should better clarify what they define as inadequate exposure

Introduction: The authors should better describe what the chemical characteristics of OPEs are, in which contexts are used, and what the sources of exposure are for the general population.

Introduction, lines 41-43: The routes of exposure to humans should be more precisely described, and more reference literature should be provided to support the claim that defines dust ingestion as the main route of exposure

Introduction, lines 66-67: The authors should better define what they mean for "health analysis"

Material and Methods, lines 80-81: A more accurate description of the rationale behind the choice of hepatocytes as reference cells for risk assessment of exposure from OPEs could improve the quality of the manuscript.

Material and Methods, line 80: A grammar check is required.

Material and Methods, lines 100-108: Authors should define more precisely which datasets were used for the literature search, all keywords used, what criteria were used for the selection of the studies considered (publication period, quality assessment). 

Discussion, lines 298-303, lines 326-329, lines 336-338, lines 340-351: Bibliographic references are missing

Discussion: Authors should add a paragraph highlighting the limitations and strengths of the study

Conclusions: the conclusions should be less generalized and refer exclusively to the results obtained, highlighting the limitations of the methods used.

Reviewer 2 Report

Authors present results of their research on elaborating model for liver-based probabilistic risk assessment of exposure to organophosphate esters via dust ingestion.

Abstract is not sufficiently informative; add digits on model data used and results obtained. Present pros and cons of Your approach. Prevailing part of introduction is trivial and too general e.g. “In recent years, the toxic effect, and mechanisms of OPEs have been reported.” – everything can be a toxin and nothing can be a toxin… or “It can cause liver vacuolation, inflammation, and apoptosis, as well  as increased liver size” – increased liver size is result of inflammation so secondary results and liver does not undergo apoptosis – liver’s cells can undergo apoptosis.

Editorial: in the entire manuscript add space in front of reference []. Put foreign words e.g. “via” with italics.

L 44-49: it is too obvious to present it here. Give more details on IVIVE approach, pros and cons, review studies using it in more details.

Fig. 1. – remove colouring of this figure.

2.1.1 – why only TPHP and TDCPP were selected? Please explain in the text

2.2 – 2.5 – nice work!

Fig 3 and 4 are of very low quality, please modify it.

L 204: add space between value and unit (also in other places)

Results are clearly presented and are scientifically sound. Well done.

Discussion is interesting and comprehensive (although English revision is necessary).
